# Trends in prostate cancer incidence and mortality to monitor control policies in a northeastern Brazilian state

Carlos Anselmo Lima[ID][1,2,3,4☯] *, Brenda Evelin Barreto da Silva[2☯], Evânia Curvelo Hora[2☯], Marcela Sampaio Lima[2,4☯], Erika de Abreu Costa Brito[2,4☯], Marceli de Oliveira Santos[5☯], Angela Maria da Silva[2,3,4☯], Marco Antonio Prado Nunes[2,3,4☯], Hugo Leite de Farias Brito[2,4☯], Marcia Maria Macedo Lima[3,4☯]

1 Aracaju Cancer Registry, Aracaju, Sergipe, Brazil, 2 Health Sciences Graduate Program, Federal University of Sergipe, Aracaju, Sergipe, Brazil, 3 Programa de Pós-graduação Profissional em Gestão e Inovação Tecnológica em Saúde, Aracaju, Sergipe, Brazil, 4 University Hospital, EBSERH, Federal University of Sergipe, Aracaju, Sergipe, Brazil, 5 CONPREV, Brazilian National Cancer Institute, Rio de Janeiro, Rio de Janeiro, Brazil

☯ These authors contributed equally to this work.
* carlos.a.lima@ufs.br

**Data Availability Statement:** All relevant data are within the manuscript and its Supporting Information files.

## Abstract

Prostate cancer differently affects different regions of the world, displaying higher rates in more developed areas. After the implementation of prostate-specific antigen (PSA) testing, several studies described rising rates globally, but it is possible that indolent lesions are being detected given the lack of changes in mortality data. The Brazilian government recommends against PSA screening in the male population regardless of age, but the Urology Society issued a report recommending that screening should start at 50 years old for certain men and for those aged ≥75 years with a life expectancy exceeding 10 years. In this study, we examined the incidence and mortality rates of invasive prostate cancer over time in the Sergipe state of Brazil. The databases of the Aracaju Cancer Registry and Mortality Information System were used to calculate age-standardized rates for all prostate tumors (International Classification of Diseases 10th edition: C61 and D07.5) in the following age ranges: 20–44, 45–54, and ≥65 years. We identified 3595 cases of cancer, 30 glandular intraepithelial high-grade lesions, and 3269 deaths. Using the Joinpoint Regression Program, we found that the incidence of prostate cancer dramatically increased over time until the mid-2000s for all age groups, after which the rates declined. Prostate cancer mortality rates increased until 2005, followed by a non-significant annual percent change of 22.0 in 2001–2005 and a stable rate thereafter. We noticed that the increases and decreases of the incidence rates of prostate cancer were associated with the screening recommendations. Meanwhile, the increased mortality rates did not appear to be associated with decreased PSA testing; instead, they were linked to the effects of age and improvements in identification of the cause of death. Thus, we do not believe a PSA screening program would benefit the population of this study.

**Funding:** CAL received a Research Development Grant from the Fundação de Apoio à Pesquisa e à Inovação Tecnológica do Estado de Sergipe – FAPITEC/SE.

**Competing interests:** The authors have declared that no competing interests exist.

## Introduction

Prostate cancer (PCa) is associated with high incidence rates in developed countries and increasing rates in developing areas, especially in those in which prostate-specific antigen (PSA) testing is available for asymptomatic men [1]. However, most detected PCa lesions are low- to intermediate-grade tumors with slow progression, and only a small percentage of cases have a more aggressive course [2–4]. For this reason, PSA screening combined with ultrasound followed by guided biopsy is more likely to detect indolent lesions, which can lead to overdiagnosis, thereby inflating incidence statistics and overestimating the number of deaths attributable to PCa [5,6]. Conversely, modern multiparametric magnetic resonance imaging (MP-MRI) fusion-guided biopsy is believed to better identify more aggressive lesions than standard techniques [7]. Corroborating this belief, the PROMIS study found that 25% of men would avoid biopsy if MP-MRI were used as triage [4].

The age-standardized rate (ASR) of PCa per 100,000 men (standardized to the global population) varies by region, usually being higher in more developed regions (Oceania, 79.1; North America, 73.7; Europe, 62.1; South America, 60.4; Africa, 26.6; Asia, 11.5). The ASRs of mortality are commonly higher in less developed areas (Africa, 14.6; South America, 14.0; Europe, 11.3; Oceania, 10.7; North America, 7.7; Asia 4.5) [8]. Estimates of the yearly cancer incidence for 2020–2022 indicate that the incidence of PCa will also vary by region in Brazil, with mean ASRs per 100,000 men of 47.8, 80.0, 75.7, 50.8, and 46.3 for the North, Northeast, Midwest, Southeast, and South regions, respectively. In addition, a rate of 122.5 has been estimated for the state of Sergipe, including a rate of 81.9 in the capital Aracaju (incidence estimates not corrected for PSA screening; thus, resulting in high figures) [9].

With the advent of PSA screening for PCa, the incidence of the malignancy has increased since the early 1990s [10,11], but given the questionable impact on mortality rates, the real-world benefit of increased screening has not been clarified. In 2002, the United States Preventive Services Task Force (USPSTF) reported that there was no evidence for or against the continued use of PSA screening [12]. However, in 2008, another report concluded that there was no benefit of screening in men older than 75 years [13], and in 2012, they recommended against PSA screening for all ages [14].

In Brazil, following the international trend, PSA testing spread as the ideal method for the early detection of PCa. Subsequently, public managers and cancer societies launched campaigns to increase awareness of the test. In 2008, the Brazilian National Cancer Institute (INCA) issued a recommendation against PSA testing on the basis of several international studies that demonstrated the lack of utility of the test. However, because of backlash from cancer societies, INCA later withdrew the recommendation. In 2010, Brazil's Ministry of Health issued a report recommending against PSA screening, which was ratified by INCA in 2013 [15].

After analyzing the benefits and drawbacks of screening subpopulations at higher risk, the USPSTF update in 2018 [16] recommended that men aged 55–69 years should discuss the risks and benefits of screening with their physicians and cautioned against screening for men aged ≥70 years. In Brazil, the Urology Society issued a report indicating that screening should begin for certain men aged ≥50 years and those aged ≥75 years with a life expectancy exceeding 10 years [17].

It is noteworthy that the decisions by public health policies to support or oppose PCa screening markedly affected the recorded incidence. However, the effects of these recommendations on mortality data are unclear, mainly because to the multiple causes of death in older men [10,11]. With the restriction of screening guidelines, the incidence of PCa has tended to decrease incidence in many regions [18]. Thus, the present study analyzed incidence and

mortality trends to provide support for health policy managers in assessing the need for revised screening policies for PCa.

## Materials and methods

The state of Sergipe is located in the Northeast Region of Brazil, and the estimated population in 2019 was 2,298,696. Aracaju, Sergipe is covered by the Aracaju Cancer Registry (CR), and the estimated population of the city in 2019 was 657,013 [19]. The Brazilian health system consists of three sectors: public, which is funded by the Universal Healthcare System (SUS); private, consisting of non-profit institutions co-funded by SUS; and a supplementary sector consisting of health insurance plans provided through privately purchased insurance or funded by companies for their employees. The majority of the population is covered by SUS, and PSA testing is provided freely under a physician's recommendation in both urban and rural areas [20].

The incidence and mortality data were obtained from the CR (1996–2015) and the Online Mortality Atlas/Mortality Information System, Brazil (1980–2018), respectively. The CR database has good quality, presenting high microscopic verification rates, low rates of death certificate-only cases, low primary site-uncertain rates, and fair mortality-to-incidence ratios. The registry has been providing data for cancer incidence in five continents, CONCORD survival studies, and INCA's cancer incidence in Brazil.

We identified cases of PCa using the International Classification of Disease, Oncology, 3rd edition (ICD-3; topographical codes, ICD-O-3 C61 and D07.5; morphological codes, 8140/3, 8000/3, 8010/3, and 8142/2) to calculate incidence rates. For mortality rates, we employed the ICD-9 classification until 1991 and ICD-10 thereafter. We calculated age-standardized rates based on the direct method using the global population [21] to allow international comparisons. Age-specific rates were also calculated for the age groups 20–44 (young adults), 45–64 (middle-aged adults) and ≥65 years (elderly) for both incidence and mortality. To calculate crude and age-specific rates, we used the population counts from the Brazilian Institute of Geography and Statistics for each 5-year age group for the state of Sergipe and city of Aracaju, and the data were expressed as the number of annual cases and deaths attributable to PCa in the specified population per 100,000 individuals at risk [22].

To assess trends, we calculated the annual percent change (APC) and average annual percent change (AAPC), as well as their confidence intervals (CIs), using the Joinpoint Regression Program version 4.8.0.1 [23]. We used the Monte Carlo simulation of the permutation test, which is more restrictive, to allow fewer joinpoints than other models of the program. The default minimum number of joinpoints was set at zero, and the maximum number was set to permit using one joinpoint for at least seven data points. These settings met the grid search method, which produces a set of all possible positions of joinpoints, all in accordance with the parameters, to find the best fit [24].

The Research Ethics Committee of the Federal University of Sergipe approved this study. All methods were developed in accordance with the relevant guidelines and regulations. Patient databases were anonymized; thus, obtaining informed consent was not possible. Consequently, as specified in Resolution number 466, December 12, 2012, of the Ministry of Health of Brazil, the ethics committee granted exemption of the requirement for informed consent.

## Results

We retrieved data for 3595 malignant neoplasms and 30 high-grade glandular intraepithelial lesions (without PCa) from the CR for the period of 1996–2015 (Table 1).

From the online Mortality Atlas of the Mortality Information System, 1218 deaths were identified for the period 1980–2018 for the CR area, and 3269 deaths were recorded for the

**Table 1. Number and percentage of incident cases of prostate neoplasms by morphology, Cancer Registry, 1996–2015.**

| Morphology | Number | % |
|---|---|---|
| **8000/3: Malignant neoplasm, NOS** | 214 | 5.9 |
| **8010/3: Carcinoma, NOS** | 3 | 0.1 |
| **8140/3: Adenocarcinoma, NOS** | 3378 | 93.2 |
| **8148/2: Glandular intraepithelial neoplasia, high grade** | 30 | 0.8 |
| **Total** | 3625 | 100 |

NOS: Not otherwise specified.

state area. Eighteen incident cases from the CR database, 9 deaths in the CR area mortality database, and 14 deaths in the state mortality database were excluded from the calculation of age-specific rates because the ages of the individuals could not be defined. Thus, Table 2 presents the numbers of cases, deaths, and percent and mean age-standardized rates for all ages and each age group.

The analysis of trends of the incidence of PCa in the specified period (Table 3 and Fig 1) revealed marked growth until 2007, including APCs of 31.6 (95% CI = 4.8–65.3) in 1996–1999 and 5.4 (95% CI = 1.4–9.5) in 1999–2017. Thereafter, we noted a decreasing trend with an APC of −5.6 (95% CI = −8.2 to −3.0) for all ages. The pattern was similar for the 45–64-year-old group, in which the incidence increased until 2008 before decreasing, and for patients aged ≥65 years, who had increasing rates until 2006 before a subsequent downward trend.

When assessing PCa deaths in the population covered by the CR (Table 3 and Fig 2), an increasing tendency was noted (AAPC = 1.2; 95% CI = 0.5–1.8), primarily because of increases in the elderly group (AAPC = 1.9; 95% CI = 1.1–2.6).

For deaths across the state (Table 3 and Fig 3), the rate tended to increase until 2001 (APC = 4.8; 95% CI = 2.8–6.9), followed by a non-significant change between 2001 and 2005 (APC = 22.0; 95% CI = −0.4–49.5) and stabilization thereafter.

As presented in Fig 4, trends in mortality related to ill-defined causes were introduced to assess the correlation with the increased mortality rate from PCa in the entire state population. The data revealed a striking decreasing trend starting in 1992 followed by stabilization at low proportions in 2006.

## Discussion

The present study analyzed the databases of the CR and the Mortality Information System on the online Mortality Atlas. After calculating age-standardized and age-specific rates, trend

**Table 2. Number and percentage of incident cases of prostate cancer and deaths by age group.**

| Age group | Inc cases | % | Rate | Death CR | % | Rate | Death ST | % | Rate |
|---|---|---|---|---|---|---|---|---|---|
| <**45** | 23 | 0.6 | 1,2 | 5 | 0.4 | 0.0 | 13 | 0.4 | 0.0 |
| **45–64** | 1215 | 34.0 | 197.6 | 165 | 13.7 | 18.0 | 381 | 11.7 | 9.9 |
| **65+** | 2339 | 65.4 | 1312.3 | 1039 | 85.9 | 393.8 | 2861 | 87.9 | 184.6 |
| **All** | 3577 | 100 | 113.8 | 1209 | 100 | 22.8 | 3255 | 100 | 11.1 |

Inc cases: Incident cases in the cancer registry area; Death CR: Deaths in the cancer registry area; Death ST: Deaths in the state population.

S1–S3 Tables present the number of annual incident cases and age-standardized rates with corresponding CIs for invasive carcinoma, CR area deaths, and state deaths. The overall data indicate year-to-year stability.

**Table 3. Joinpoint analyses of prostate cancer incidence and mortality trends.**

|  | Inc | CR |  |  | Mor | CR |  |  | Mor | ST |  |  |
|---|---|---|---|---|---|---|---|---|---|---|---|---|
| Age group | JP Seg | APC | (95% CI) | p | JP Seg | APC | (95% CI) | p | JP Seg | APC | (95% CI) | p |
| <45 | NF | NF | NF | NF | NF | NF | NF | NF | NF | NF | NF | NF |
| **45–64** | 1996–2008 | 11.1* | (7.9; 14.3) | <0.001 | 1980–2018 | −1.2 | (−2.9; 0.6) | 0.187 | 1980–2018 | 2.4* | (1.0; 3.9) | 0.002 |
|  | 2008–2015 | −8.2* | (−11.8; −4.4) | <0.001 |  |  |  |  |  |  |  |  |
| **≥65** | 1996–2005 | 9.3* | (3.6; 15.2) | 0.003 | 1980–2018 | 1.9* | (1.1; 2.6) | <0.001 | 1980–2001 | 4.8* | (2.8; 6.9) | <0.001 |
|  | 2005–2015 | −5.6* | (−8.7; −2.5) | 0.002 |  |  |  |  | 2001–2005 | 26.6* | (5.0; 52.6) | 0.015 |
|  |  |  |  |  |  |  |  |  | 2005–2018 | 1.1 | (−0.4; 2.6) | 0.153 |
| **All** | 1996–1999 | 31.6* | (4.8; 65.3) | 0.022 | 1980–2018 | 1.2* | (0.5; 1.8) | 0.001 | 1980–2001 | 4.9* | (2.7; 7.1) | <0.001 |
|  | 1999–2007 | 5.4* | (1.4; 9.5) | 0.012 |  |  |  |  | 2001–2005 | 22.0 | (−0.4; 49.5) | 0.055 |
|  | 2007–2015 | −5.6* | (−8.2; −3.0) | 0.001 |  |  |  |  | 2005–2018 | 0.3 | (−1.3; 1.9) | 0.698 |

Inc CR: Incidence in the cancer registry area; Mor CR: Mortality in the cancer registry area; Mor ST: Mortality in the state population; JP Seg: Time range segment; APC: Annual percent change; CI: Confidence interval; NF: Model not fitted.

*Significant APC, p ≤ 0.05.

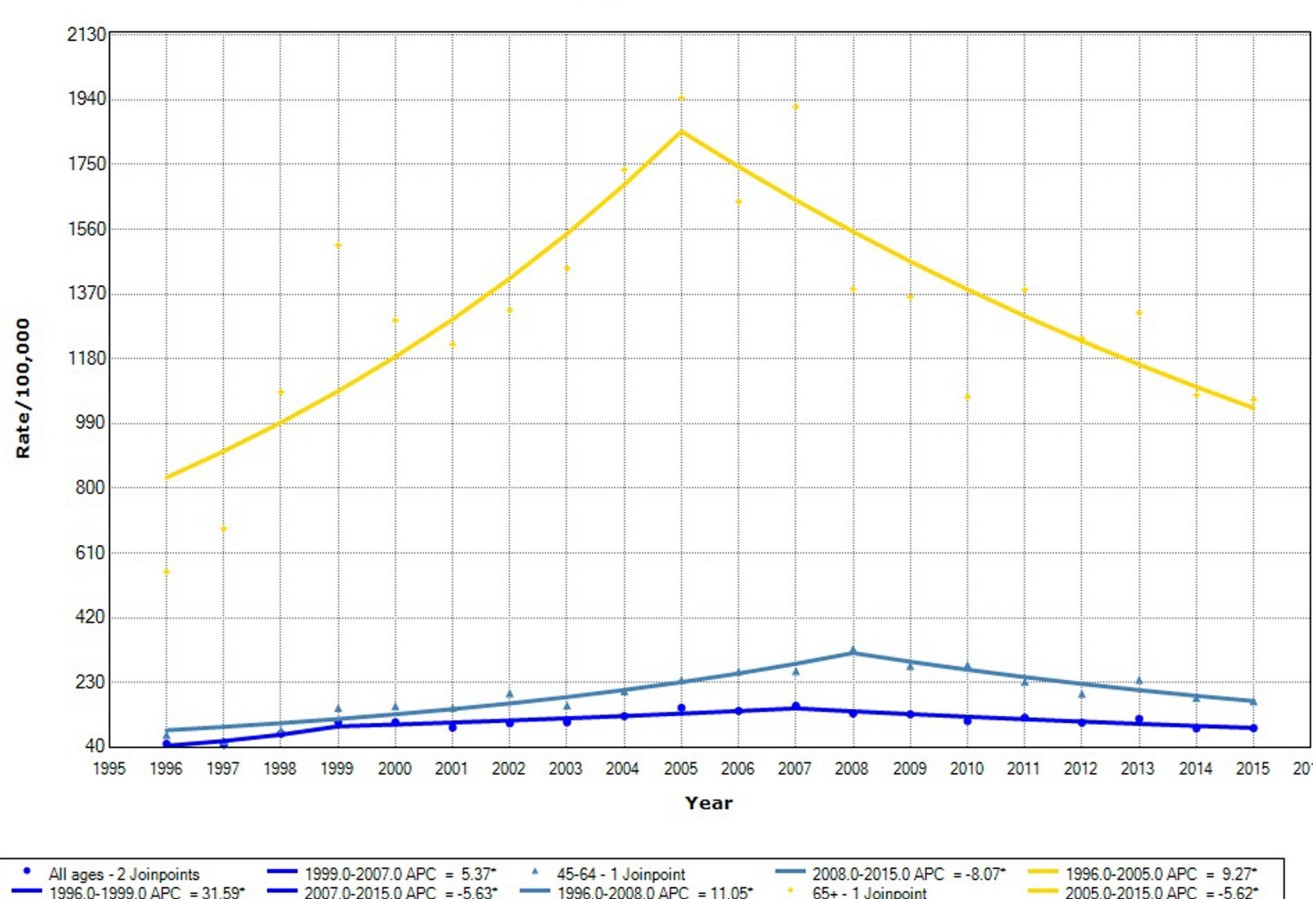

**Fig 1. Incidence trends for prostate cancer in the cancer registry area, 1996–2015.**

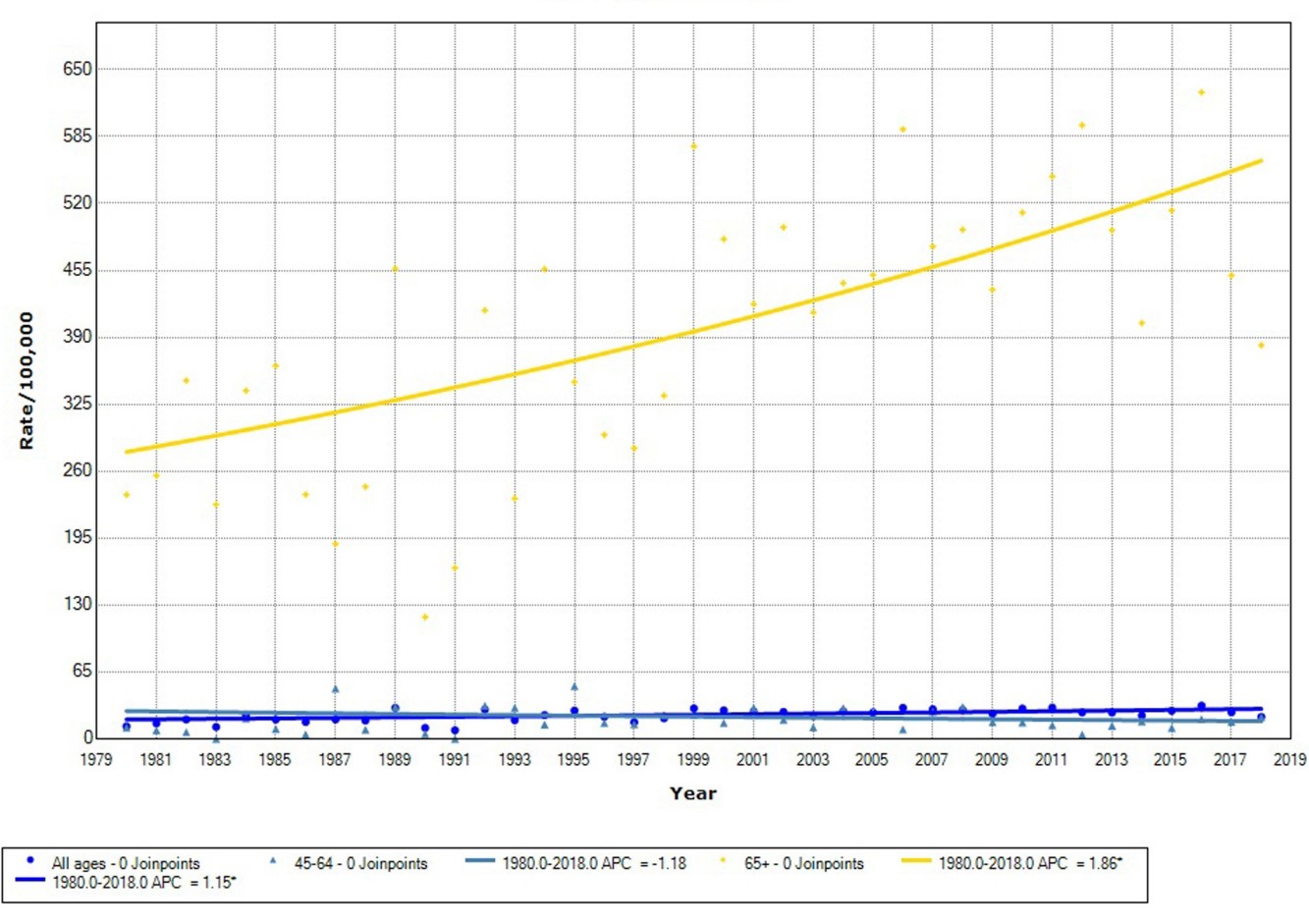

**Fig 2. Mortality trends for prostate cancer in the cancer registry area, 1980–2018.**

analyses were performed. Regarding incidence, there was a steady increase until 2007, after which a decrease was noted, although rates remained higher than those at the beginning of the time series. The trend curve was markedly influenced by the elderly age group. Mortality rates in the CR area tended to increase over the entire time series, especially starting in 1996, and more pronounced increases were noted in patients aged ≥65 years. The mortality trend for the entire state was similar, increasing mainly starting in the mid-1990s, and higher rates were recorded for the ≥65 age group, albeit with much lower rates than those in the CR area. We also observed that the mortality rates in the CR area were approximately 2-fold higher than those in the state.

In a systematic review, Dasgupta *et al.* found that PCa indicators varied geographically in areas with different screening policies [25]. In Brazil, efforts to increase PSA screening were exerted, including the utilization of mobile testing units [26]. Conversely, Araujo *et al.* observed a reduction of PSA testing rates starting in 2009 in private laboratories but not among men aged >74 [27]. Earlier observational studies demonstrated that the increased number of biopsies and the diagnosis of indolent tumors did not yield benefits, and the impact on mortality was questionable, leading to unnecessary treatment risks, such as impotence and urinary complications [28–31]. Consequently, several countries issued policies recommending

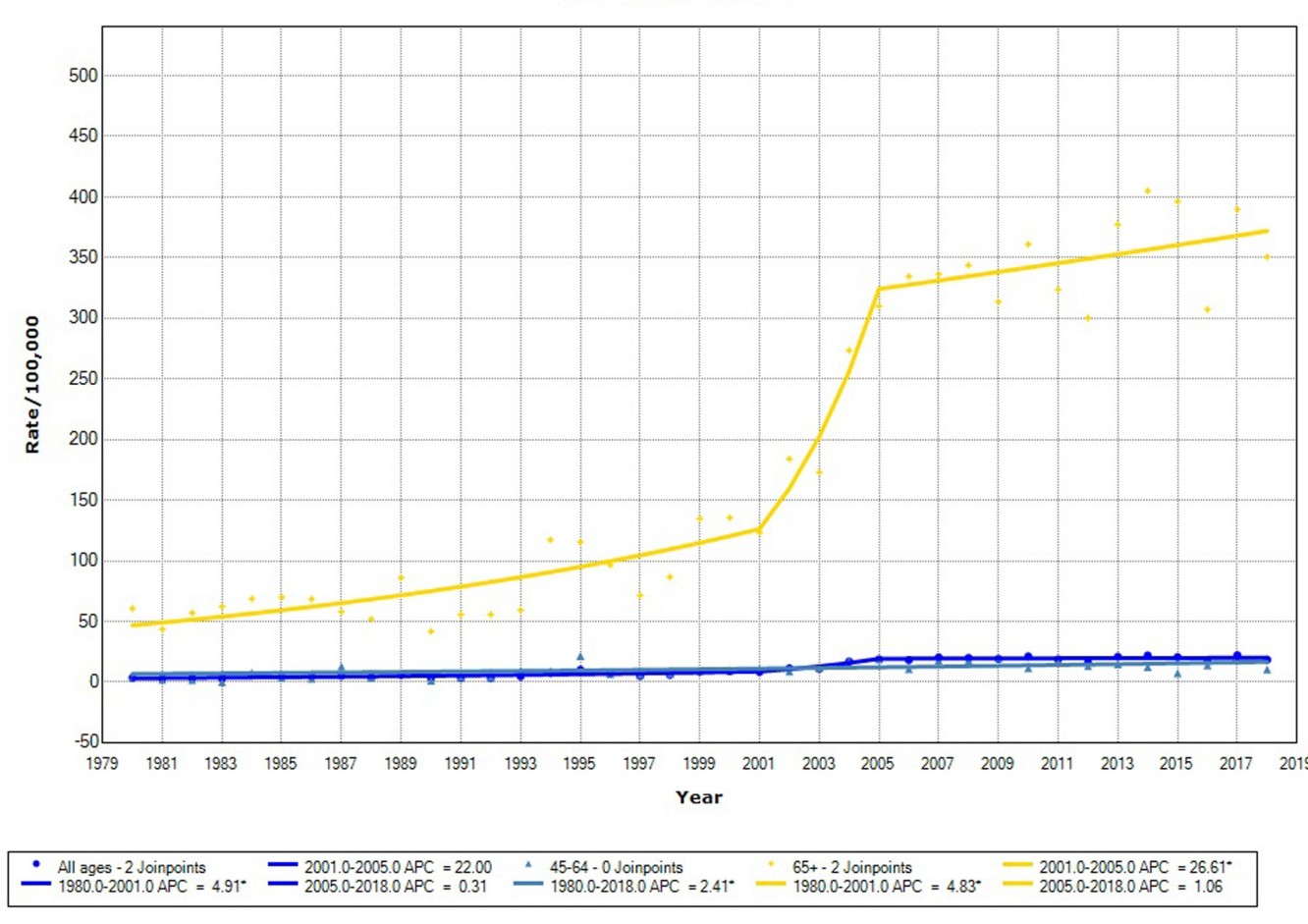

**Fig 3. Mortality trends for prostate cancer in the state of Sergipe, 1980–2018.**

against PSA testing, including Brazil [15]. Thus, the incidence rates of PCa have declined, but they remain higher than those in the pre-PSA period [18,32], as also revealed in the present study. Conversely, there has been an uptick in the incidence in geographic areas with low rates [33].

The utility of PSA screening has not been clarified. Some studies suggest that there may be differences in the incidence and mortality associated with genetic and racial factors [34]. Thus, the need of screening could be based on individual risk, considering age, race, and family history. Because the risk of PCa is directly proportional to age, for individuals younger than 60, screening is only beneficial among men with a family history of PCa. The family history should include multiple relatives with PCa, especially at younger ages, as well as relatives diagnosed with advanced PCa who developed metastases or died of PCa [16]. People of African descent in the United States have a higher incidence of PCa and a worse prognosis after diagnosis [34]. However, in Brazil, clear racial separation is not often observed [34], in line with the study findings.

According to Sierra *et al.* [35], the geographic and temporal variation in the incidence of PCa in South and Central America may be attributable to differences in diagnostic techniques of diagnosis and cancer registration, access to health services, the completion of death

**Multiple Joinpoint Models**

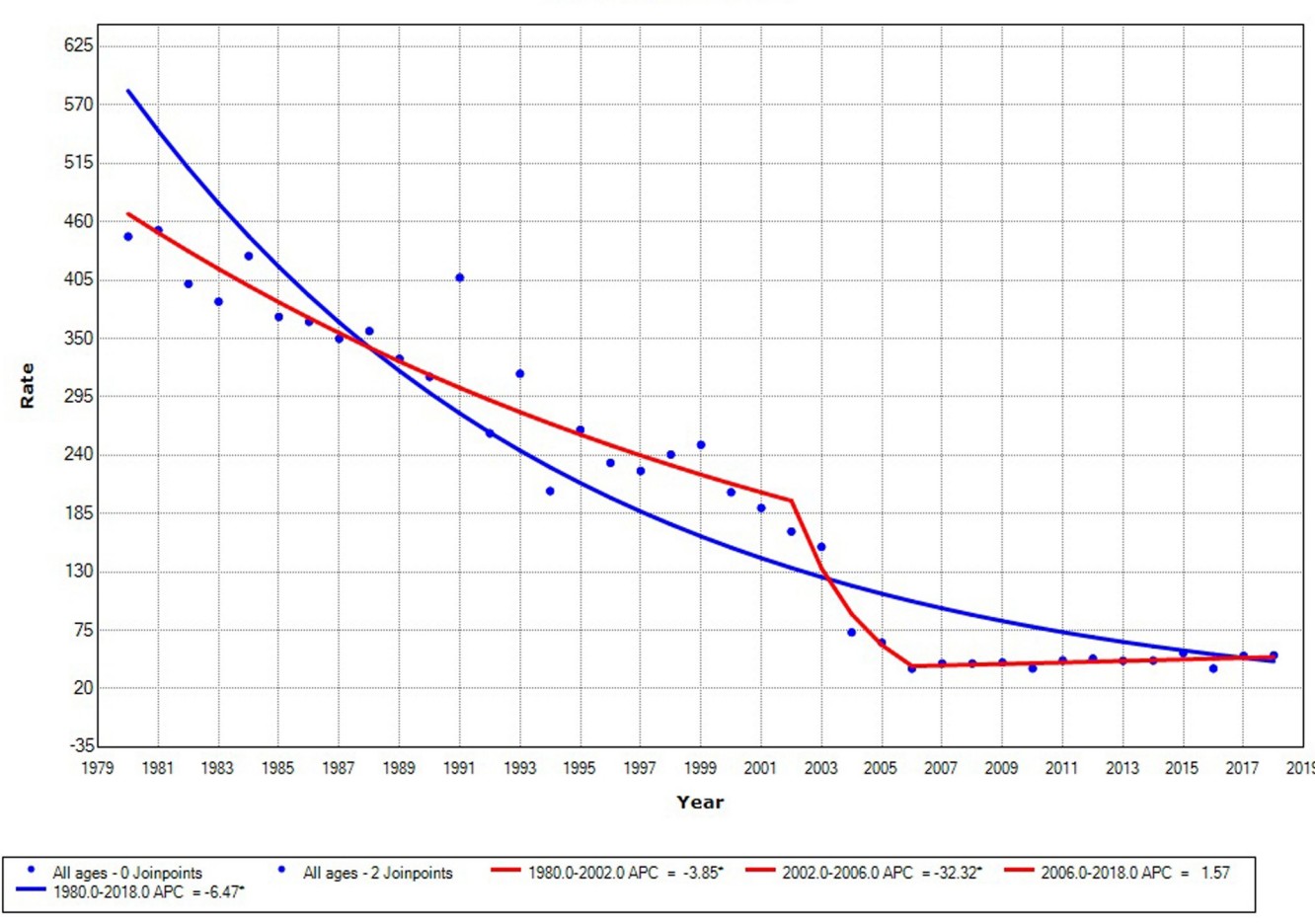

**Fig 4. Mortality trends by ill-defined causes in males in Sergipe, 1980–2018.**

certificates, and public awareness. Considering the aforementioned factors and the indolent course of PCa in most cases, screening practices with PSA, ultrasound, and even MRI can lead to the unnecessary diagnosis of many tumors without improving mortality rates. These strategies can also increase the diagnosis of cancer in young patients, leading to radical treatment and the use of additional health resources [36,37]. Indeed, Gray *et al.* [38] demonstrated that 80% of diagnoses involved low- to intermediate-risk tumors, but the rate of radical prostatectomy has increased among all risk groups.

Trend analyses in developed countries have revealed consistent decreases in incidence and mortality rates [18,30], and the screening protocols did not negatively affect the evolution of patients. Braga *et al.* [39] observed an increase in mortality rates in Brazil, especially in the Northeast Region, and concluded that the finding was attributable to age and that the differences between Brazilian regions were possibly related to inequalities in health service access and use. We hypothesized that the increased mortality rates recorded in our study are linked to both age and improvements in the certification of cause of death, as confirmed by the decreased rate of mortality from ill-defined causes, in line with previous findings [40]. The decreased mortality rates in developed countries may have been associated with improvements in treatment options. Conversely, the increased rates in many Asian and South American

countries, as well as certain Eastern European countries, may have been attributable to other factors such as obesity and unhealthy diets [8]. We understand that the increased mortality rate in our area has another contributing factor, namely the attribution of PCa as the cause of death, consequent to the increase in the diagnosis of indolent lesions via PSA testing and the increased number of biopsies, a fact also referenced by Culp *et al.* [8].

The strengths of the study include the use of data from long time series. Concerning the incidence data, we recorded the following quality indicators: microscopic verification, 94.2%; death certification-only, 4.5%; and unknown primary site, 0.6% (S4 Table). In addition, a high rate of adenocarcinoma was noted in histopathological records.

Concerning the study limitations, there were some uncertainties in the official mortality data, such as the high number of ill-defined cases in the initial years of the series, which jeopardized the interpretation of the study results. By correcting the causes for competitive risks, bias could possibly be avoided, and comparisons with net survival studies could be initiated. Regarding the incidence data, it was not possible to perform the calculation by cancer stage or tumor aggressiveness as defined using the Gleason score or new grading systems.

## Conclusions

Our findings suggest that the observed trends in the incidence rates of PCa in the study population might have resulted from changes in screening recommendations. Concerning the increased mortality rates, we believe that these findings are attributable to age and improvements in certification of the cause of death, as highlighted by the reduced rates of death from ill-defined causes. Therefore, on the basis of the establishment of vital statistics, the late age at diagnosis, and the indolent course of the disease, we could infer that many individuals died from causes other than PCa. Additional research might be necessary, for instance, to investigate whether radical treatment in patients with aggressive PCa would reduce mortality rates or whether any causative agent would explain the high incidence rates in the study population.

On the basis of these findings, we would not recommend the implementation of systematic PCa screening policies to health policy managers. Conversely, we agree with the need for targeted public awareness campaigns calling attention to preventative strategies and highlighting risk factors for PCa. In addition, individuals aged 55–69 should acquire information about their personal risks and benefits to ensure individualized decision-making with the help of a specialized professional.

## Supporting information

**S1 Table. Number of incident cases, age-standardized rates and confidence intervals, cancer registry area (CR).**
(PDF)

**S2 Table. Number of deaths, age-standardized rates and confidence intervals, cancer registry area (CR).**
(PDF)

**S3 Table. Number of deaths, age-standardized rates and confidence intervals, state area (ST).**
(PDF)

**S4 Table. Data quality of incidence data, Aracaju Cancer Registry, 1996–2015.**
(PDF)

## Acknowledgments

We thank the following personnel of the Cancer Registry for their work in collecting data and preparing the database for this research: José Erinaldo Lobo de Oliveira, Elma Santana de Oliveira, Maria das Graças Prata França, Sueli Pina Vieira, Marina Kobilsek, Maria Cristina Conceição Coelho Santos, Maria das Graças Rodrigues de Melo, and Cecília Ferreira. We thank Joe Barber Jr., PhD, from Edanz Group (https://en-author-services.edanz.com/ac) for editing a draft of this manuscript.

## Author Contributions

**Conceptualization:** Carlos Anselmo Lima, Marcela Sampaio Lima, Marceli de Oliveira Santos.

**Formal analysis:** Carlos Anselmo Lima, Marcela Sampaio Lima.

**Methodology:** Carlos Anselmo Lima, Marceli de Oliveira Santos.

**Validation:** Brenda Evelin Barreto da Silva, Evânia Curvelo Hora, Erika de Abreu Costa Brito, Angela Maria da Silva, Marco Antonio Prado Nunes, Hugo Leite de Farias Brito, Marcia Maria Macedo Lima.

**Writing – original draft:** Carlos Anselmo Lima.

**Writing – review & editing:** Carlos Anselmo Lima, Brenda Evelin Barreto da Silva, Evânia Curvelo Hora, Erika de Abreu Costa Brito, Marceli de Oliveira Santos, Angela Maria da Silva, Marco Antonio Prado Nunes, Hugo Leite de Farias Brito, Marcia Maria Macedo Lima.

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
