## [Decision Letter · Decision Letter 0]

9 Dec 2020

PONE-D-20-33241

Trends in prostate cancer incidence and mortality to monitor control policies in a northeastern Brazilian state

PLOS ONE

Dear Dr. Lima,

Thank you for submitting your manuscript to PLOS ONE. After careful consideration, we feel that it has merit but does not fully meet PLOS ONE’s publication criteria as it currently stands. Therefore, we invite you to submit a revised version of the manuscript that addresses the points raised during the review process.

We look forward to receiving your revised manuscript.

Kind regards,

Mieke Van Hemelrijck

Academic Editor

PLOS ONE

Journal Requirements:

2) Please include captions for your Supporting Information files at the end of your manuscript, and update any in-text citations to match accordingly. Please see our Supporting Information guidelines for more information: http://journals.plos.org/plosone/s/supporting-information.

Reviewers' comments:

Reviewer's Responses to Questions

**Comments to the Author**

1. Is the manuscript technically sound, and do the data support the conclusions?

Reviewer #1: Partly

Reviewer #2: Yes

2. Has the statistical analysis been performed appropriately and rigorously? 

Reviewer #1: I Don't Know

Reviewer #2: Yes

3. Have the authors made all data underlying the findings in their manuscript fully available?

Reviewer #1: Yes

Reviewer #2: Yes

4. Is the manuscript presented in an intelligible fashion and written in standard English?

Reviewer #1: No

Reviewer #2: Yes

5. Review Comments to the Author

Reviewer #1: Manuscript review PONE-20-33241

I acknowledge the importance of reporting trends in prostate cancer incidence and mortality from developing countries and commend the authors for preparing this manuscript. Some minor editing of the English would improve this manuscript.

My major concern with the manuscript is the strong emphasis on interpreting trends in PCa incidence in relation to PSA screening policies without providing any strong evidence for this, and no specific data on PSA testing.

In the introduction, the authors refer to the ‘implementation’ of screening with PSA. In most developed countries no PSA screening programs have been implemented and PSA screening is not recommended. It might be better to specifically refer to the USA policies (particularly if Brazilian policies were strongly influenced by policies in the USA) or to refer to increased availability and uptake of PSA testing in asymptomatic men if wanting to describe practices more globally.

To strengthen their argument that PCa trends in their region are tied to PSA screening policies (and uptake) in their country, I suggest the authors provide more detail about specific policies and if available add data on PSA testing during the study period. The authors only make reference to government recommendations not to undergo PSA testing in 2010, which was ratified in 2013. I feel more detail for the period prior to 2010 is warranted. Expanding on references 13 and 23 would be helpful and may further support their argument. (I note ref 23 notes a decline in PSA testing since 2001 which is somewhat in conflict with the conclusions made by authors of the current manuscript.)

More detail about the health system in Sergipe/Brazil to provide context to the international readership in relation health care access, i.e. Is health care (including asymptomatic PSA testing) freely available to all, or do patients pay a fee for service? How much variation in access in rural vs metropolitan areas?

The authors do not provide very much information on the quality of the Aracaju registry. Estimates of the coverage, completeness and timeliness of caner data collection would be useful. I not the include of data to the end of 2019. This suggests confidence on behalf of the authors that data on cancer incidence is complete for that year. Many very well-resourced registries would not include data up to such a recent period as it would not be considered complete or have been checked for quality/accuracy within such a short time.

Methods: No information is provided as to the source of the denominator population used in determining age specific rates. Was any account taken for changing population age structure in the region over the time period? The age ranges used in this study seem very broad (~20year intervals). If true, this suggests a lack of precision in calculating ASRs. Can the methodology be expanded/clarified further?

I am concerned about projections of incidence rates to the whole state based on cancer data for less than one quarter of the population. The authors indicate that no account was made for PSA screening bias, even though they are claiming throughout the paper that PSA screening rates are driving the incidence rates. Does this method account for difference in the age structure for Aracaju and the rest of the state?

Having stated that incidence rates were projected for the whole state, it is clear which if any tables and figures are presenting data for the region covered by the registry and for the whole state. (I could not see any reference to incidence for the whole state in any table/ figure headings nor in the suppl. material).

Conclusion

I do not feel it is appropriate to conclude that indolent cancers account for 80% of PCa cases based on what is presented in the current study. This was not a direct question investigated in this study. The claim also ignores the impact of radical treatments in patients with significant PCa, whose risk of PCa death would be substantially reduced by having undergone treatment.

The conclusion of this study is heavily focussed toward drawing links between PCa incidence trends and PSA testing policies, and this explanation is strongly stated strongly in the concluding paragraph. I don’t feel that the study presents enough direct evidence to support this conclusion.

Minor points:

The sentences last 2 sentences in the first paragraph leads to the interpretation that authors are saying MRI guided biopsies lead to overdiagnosis. I feel that the evidence presented in ref 3 does not indicate this. I suggest editing/reordering these sentences.

In paragraph 2 of the introduction are the ASRs quoted for global regions and for states within brazil using the same reference populations (ie world pop)? The figures quoted suggest that PCa incidence rates in Sergipe are considerably higher than most developed countries including the USA.

I am not convinced about arguments that increases in mortality rates are due to aging population. This should not be the case if age standardisation is undertaken to produce the mortality rates.

Reviewer #2: The manuscript examined the incidence and mortality rate of prostate cancer in a Brazilian state, Sergipe over a 20 years period in order to inform stakeholders of the potential benefits/disadvantages of a prostate cancer screening program.

The abstract has some typos and an acronym not defined previously (APC), please amend that. The authors do not conclude in the abstract whether a screening program could be beneficial or not in that region. I will suggest to include a conclusion to respond to this question introduced in the aim of the study.

Introduction - please add a reference to PROMIS study in the diagnostics section. The following sentence will need to be justified with further references "has shown a slow progression in almost 75% percent of cases and the remaining can behave

more aggressively." The sentence "Mortality ASRs are higher in less developed areas" cannot be justified with the ASRs presented from the different continents being Europe the 2 continent with a higher mortality rate just after Africa. Please amend it accordingly.

The methods and results are well described, some comments below.

Please be careful in the table captions to describe all the variables included in the tables and consistent with the punctuations in the tables (e.g.: 4.9*(2.7;7.1) or 0.3(-1.3;1.9)) Moreover it could be formulated as rate (xx, xx). Please add an extra column for the p values as it is a bit unclear in the current format. Please include figures with higher definition as the current figures are blur.

Discussion

Please include more studies justifying the following sentence -"Earlier observational studies began to demonstrate that the increased amount of biopsies and the diagnosis of indolent tumors did not yield benefits, and the impact on mortality

was questionable;"

"Some suggest that factors there may be differences in the incidence and mortality produced by genetic and racial"- studies in first and second generation of emigrant populations suggested that diet and other life style factors may have a role in the increase rates of PCa incidence observed in second generations (https://seer.cancer.gov/archive/studies/surveillance/study5.html). Could these other potential factors have a role in the patterns of incidence observed in Brazil?

Conclusion, just a question for the authors, do you consider that the rise and fall in the

incidence rates of PCa are solely a result from the temporal forms of recommendation for screening?

Overall the study is well conducted and well described.

6. PLOS authors have the option to publish the peer review history of their article (what does this mean?). If published, this will include your full peer review and any attached files.

Reviewer #1: No

Reviewer #2: **Yes: **Aida Santaolalla

---

## [Author Response · Author response to Decision Letter 0]

23 Jan 2021

Response to reviewers

Reviewer #1: Manuscript review PONE-20-33241

I acknowledge the importance of reporting trends in prostate cancer incidence and mortality from developing countries and commend the authors for preparing this manuscript. Some minor editing of the English would improve this manuscript.

R. English editing was done, and we thank Joe Barber Jr., PhD, from Edanz Group (https://en-author-services.edanz.com/ac) for editing a draft of this manuscript. 

My major concern with the manuscript is the strong emphasis on interpreting trends in PCa incidence in relation to PSA screening policies without providing any strong evidence for this, and no specific data on PSA testing.

R. We understand the concern and made some changes in the manuscript to minimize emphasis on interpretating trends in relation to PSA Screening policies.

In the introduction, the authors refer to the ‘implementation’ of screening with PSA. In most developed countries no PSA screening programs have been implemented and PSA screening is not recommended. It might be better to specifically refer to the USA policies (particularly if Brazilian policies were strongly influenced by policies in the USA) or to refer to increased availability and uptake of PSA testing in asymptomatic men if wanting to describe practices more globally.

R. We accepted the suggestion and made changes in the manuscript.

To strengthen their argument that PCa trends in their region are tied to PSA screening policies (and uptake) in their country, I suggest the authors provide more detail about specific policies and if available add data on PSA testing during the study period. The authors only make reference to government recommendations not to undergo PSA testing in 2010, which was ratified in 2013. I feel more detail for the period prior to 2010 is warranted. Expanding on references 13 and 23 would be helpful and may further support their argument. (I note ref 23 notes a decline in PSA testing since 2001 which is somewhat in conflict with the conclusions made by authors of the current manuscript.)

R. We included more detail concerning references 13 (now 15) and 23 (now 26) aiming to strengthen our argument and minimize emphasis on PSA screening policies.

More detail about the health system in Sergipe/Brazil to provide context to the international readership in relation health care access, i.e. Is health care (including asymptomatic PSA testing) freely available to all, or do patients pay a fee for service? How much variation in access in rural vs metropolitan areas?

R. We included Information about the Brazilian Health System.

The authors do not provide very much information on the quality of the Aracaju registry. Estimates of the coverage, completeness and timeliness of caner data collection would be useful. I not the include of data to the end of 2019. This suggests confidence on behalf of the authors that data on cancer incidence is complete for that year. Many very well-resourced registries would not include data up to such a recent period as it would not be considered complete or have been checked for quality/accuracy within such a short time.

R. We included Information on the quality of data of Aracaju Cancer Registry. About data for 2019: our purpose was to provide current incidence estimates for the area of study; however, the suggestion to exclude that Information was accepted.

Methods: No information is provided as to the source of the denominator population used in determining age specific rates. Was any account taken for changing population age structure in the region over the time period? The age ranges used in this study seem very broad (~20year intervals). If true, this suggests a lack of precision in calculating ASRs. Can the methodology be expanded/clarified further?

R. To calculate age-specific rates, we used counts from censuses and population estimates for each the years in-between provided by The Brazilian Institute of Geography and Statistics for the state of Sergipe and the capital Aracaju (1980-2018); and those rates expressed the number of annual deaths of PCa in the specified Population per 100,000 individuals at risk. That was included in the manuscript.

I am concerned about projections of incidence rates to the whole state based on cancer data for less than one quarter of the population. The authors indicate that no account was made for PSA screening bias, even though they are claiming throughout the paper that PSA screening rates are driving the incidence rates. Does this method account for difference in the age structure for Aracaju and the rest of the state?

R. Actually, we used data from the cancer registry area to estimate incidence of an area without cancer registry by the method proposed by Black et al. We used official Population counts for each area, accounting for age structures for the different regions. However, as suggested we decided to retrieve that Information.

Having stated that incidence rates were projected for the whole state, it is clear which if any tables and figures are presenting data for the region covered by the registry and for the whole state. (I could not see any reference to incidence for the whole state in any table/ figure headings nor in the suppl. material).

R. It´s true we don´t have annual incidence for the whole state, and only for the cancer registry Population. That is why we had used estimates for 2019.

Conclusion

I do not feel it is appropriate to conclude that indolent cancers account for 80% of PCa cases based on what is presented in the current study. This was not a direct question investigated in this study. The claim also ignores the impact of radical treatments in patients with significant PCa, whose risk of PCa death would be substantially reduced by having undergone treatment.

R. We made changes in the manuscript to comply with this concern.

The conclusion of this study is heavily focused toward drawing links between PCa incidence trends and PSA testing policies, and this explanation is strongly stated strongly in the concluding paragraph. I don’t feel that the study presents enough direct evidence to support this conclusion.

R. We made changes to also comply with the concern.

Minor points:

The sentences last 2 sentences in the first paragraph leads to the interpretation that authors are saying MRI guided biopsies lead to overdiagnosis. I feel that the evidence presented in ref 3 does not indicate this. I suggest editing/reordering these sentences.

R. We modified in the manuscript.

In paragraph 2 of the introduction are the ASRs quoted for global regions and for states within brazil using the same reference populations (i.e. world pop)? The figures quoted suggest that PCa incidence rates in Sergipe are considerably higher than most developed countries including the USA.

I am not convinced about arguments that increases in mortality rates are due to aging population. This should not be the case if age standardization is undertaken to produce the mortality rates.

R. The rates we standardized by the world Population (Segi, 1960; Doll et al, 1966). Prostate cancer rates were presented without correction by PSA uptake. These are indeed high rates, which is a major concern for us; and, we have been discussing that with state policy makers.

Reviewer #2: The manuscript examined the incidence and mortality rate of prostate cancer in a Brazilian state, Sergipe over a 20 years period in order to inform stakeholders of the potential benefits/disadvantages of a prostate cancer screening program.

The abstract has some typos and an acronym not defined previously (APC), please amend that. The authors do not conclude in the abstract whether a screening program could be beneficial or not in that region. I will suggest to include a conclusion to respond to this question introduced in the aim of the study.

R. We amended typos and definitions of acronyms; and also, we inserted a paragraph to respond to the question of the study

Introduction - please add a reference to PROMIS study in the diagnostics section. 

R. The reference for the PROMIS study was inserted in the manuscript.

The following sentence will need to be justified with further references "has shown a slow progression in almost 75% percent of cases and the remaining can behave more aggressively." 

R. That sentence was rewritten and other references were added.

The sentence "Mortality ASRs are higher in less developed areas" cannot be justified with the ASRs presented from the different continents being Europe the 2 continent with a higher mortality rate just after Africa. Please amend it accordingly.

R. That sentence was rewritten.

The methods and results are well described, some comments below.

Please be careful in the table captions to describe all the variables included in the tables and consistent with the punctuations in the tables (e.g.: 4.9*(2.7;7.1) or 0.3(-1.3;1.9)) Moreover it could be formulated as rate (xx, xx). Please add an extra column for the p values as it is a bit unclear in the current format. Please include figures with higher definition as the current figures are blur.

R. That was corrected; we added columns for the p values. 

Discussion

Please include more studies justifying the following sentence -"Earlier observational studies began to demonstrate that the increased amount of biopsies and the diagnosis of indolent tumors did not yield benefits, and the impact on mortality was questionable;"

R. We added two more studies.

"Some suggest that factors there may be differences in the incidence and mortality produced by genetic and racial"- studies in first and second generation of emigrant populations suggested that diet and other life style factors may have a role in the increase rates of PCa incidence observed in second generations (https://seer.cancer.gov/archive/studies/surveillance/study5.html). Could these other potential factors have a role in the patterns of incidence observed in Brazil?

R. It may be so in some Brazilian states, such as São Paulo, Paraná, Santa Catarina and Rio Grande do Sul, where immigration was more pronounced ; however, that is not clear in the study population which does not hold an immigration burden.

Conclusion, just a question for the authors, do you consider that the rise and fall in the incidence rates of PCa are solely a result from the temporal forms of recommendation for screening?

R. That was initially our hypothesis, but I agree that not so much emphasis should we put on that. Of course, additional research might follow to unveil whether other causes play any role. We decided to minimize emphasis on that.

Overall the study is well conducted and well described.

---

## [Decision Letter · Decision Letter 1]

10 Mar 2021

Trends in prostate cancer incidence and mortality to monitor control policies in a northeastern Brazilian state

PONE-D-20-33241R1

Dear Dr. Carlos Anselmo Lima,

We’re pleased to inform you that your manuscript has been judged scientifically suitable for publication and will be formally accepted for publication once it meets all outstanding technical requirements.

Kind regards,

Wen-Wei Sung, M.D., Ph.D.

Academic Editor

PLOS ONE

Reviewers' comments:

Reviewer's Responses to Questions

**Comments to the Author**

1. If the authors have adequately addressed your comments raised in a previous round of review and you feel that this manuscript is now acceptable for publication, you may indicate that here to bypass the “Comments to the Author” section, enter your conflict of interest statement in the “Confidential to Editor” section, and submit your "Accept" recommendation.

Reviewer #2: All comments have been addressed

2. Is the manuscript technically sound, and do the data support the conclusions?

Reviewer #2: Yes

3. Has the statistical analysis been performed appropriately and rigorously? 

Reviewer #2: Yes

4. Have the authors made all data underlying the findings in their manuscript fully available?

Reviewer #2: Yes

5. Is the manuscript presented in an intelligible fashion and written in standard English?

Reviewer #2: Yes

6. Review Comments to the Author

Reviewer #2: (No Response)

7. PLOS authors have the option to publish the peer review history of their article (what does this mean?). If published, this will include your full peer review and any attached files.

Reviewer #2: No

---

## [Editor Report · Acceptance letter]

15 Mar 2021

PONE-D-20-33241R1 

Trends in prostate cancer incidence and mortality to monitor control policies in a northeastern Brazilian state 

Dear Dr. Lima:

I'm pleased to inform you that your manuscript has been deemed suitable for publication in PLOS ONE. Congratulations! Your manuscript is now with our production department. 

Kind regards, 

on behalf of

Dr. Wen-Wei Sung 

Academic Editor

PLOS ONE